# Effects of the In Ovo Vaccination of the ts-11 Strain of *Mycoplasma gallisepticum* in Layer Embryos and Posthatch Chicks [note 1]

**DOI:** 10.3390/ani12091120

**Published:** 2022-04-27

**Authors:** Abdulmohsen H. Alqhtani, Seyed A. Fatemi, Katie E. C. Elliott, Scott L. Branton, Jeff D. Evans, Spencer A. Leigh, Patrick D. Gerard, Edgar D. Peebles

**Affiliations:** 1Department of Poultry Science, Mississippi State University, Starkville, MS 39762, USA; aha107@msstate.edu (A.H.A.); katie.elliott@usda.gov (K.E.C.E.); 2Department of Animal Production, College of Agriculture and Food Sciences, King Saud University, P.O. Box 2460, Riyadh 11451, Saudi Arabia; 3Poultry Research Unit, USDA-ARS, Starkville, MS 39762, USA; cdbimages@bellsouth.net (S.L.B.); jeff.evans@ars.usda.gov (J.D.E.); spencer.leigh@usda.gov (S.A.L.); 4School of Mathematical and Statistical Sciences, Clemson University, Clemson, SC 29634, USA; pgerard@clemson.edu

**Keywords:** in ovo, layer embryo, *Mycoplasma gallisepticum*, transmission, ts-11-strain

## Abstract

**Simple Summary:**

Mycoplasma gallisepticum (MG) is responsible for reductions in egg production and other economic losses in the poultry industry. In this study, the potential application of in ovo vaccination of the ts-11of MG vaccine (ts-11MGV) in layer embryos for the subsequent early protection as well as live performance of pullets were investigated. The use of various dosages of live attenuated ts-11MGV ranging from 3.63 × 10^1^ to 3.63 × 10^4^ cfu that were delivered in ovo at 18 days of incubation were evaluated. The results of current study revealed that the in ovo injection of various dosage of ts-11MGV had no negative impacts on any hatch variables. Additionally, the higher dosage of ts-11MGV (3.63 × 10^4^) resulted in a reduction in body weight gain in three-week-old pullets in comparison to all other treatments. Furthermore, MG DNA remained undetectable for hatchling and three-week-old pullets and no serological response was observed at 3 weeks posthatch. Total flock protection from field-strain MG infections is facilitated by the prior systemic establishment of vaccine strains in pullets. Therefore, it is concluded that the ts-11MGV may not be an appropriate candidate for in ovo injection due to the lack of its presence in hatchlings and posthatch chicks subsequent to its in ovo administration.

**Abstract:**

The transmission of the ts-11 strain of *Mycoplasma gallisepticum* (MG) vaccine (ts-11MGV) between incubated eggs and between hatchlings that was administrated via in ovo injection, and its subsequent effects on their posthatch performance were evaluated. Marek’s disease diluent alone (sham-injected) or containing either 3.63 × 10^1^, 10^2^, 10^3^, or 10^4^ cfu of ts-11MGV was manually in ovo-injected into the amnion on 18 days of incubation. Egg residue analysis, percentage incubational egg weight loss, hatchability of viable injected eggs, and hatchling body weight (BW) were assessed. Selected hatchlings from each treatment replicate group were swabbed in the choanal cleft for MG DNA detection. Female chick live performance was also assessed through 21 days of posthatch age. Unexposed control sentinel chicks were allocated to each treatment replicate group to assess horizontal transmission. Birds were later swabbed and bled respectively, for detection of MG DNA and IgM production at 21 days posthatch. In all birds, no MG DNA was detected and SPA tests for IgM were negative. Among all variables, only 0 to 21 day BW gain was significantly affected by treatment and was lower in the 3.63 × 10^4^ ts-11 MGV treatment in comparison to all the other treatments. Because ts-11MGV does not exhibit vertical or horizontal transmission capabilities under commercial conditions, it may not be a good candidate for in ovo injection.

## 1. Introduction

*Mycoplasma gallisepticum* (MG) is highly pathogenic and causes infections in poultry that result in significant economic losses in commercial poultry operations worldwide [1,2,3,4]. An MG infection causes avian respiratory mycoplasmosis, which can lead to chronic respiratory disease in chickens and infectious sinusitis of turkeys, with subsequent increases in mortality and reduced growth rates [3,4]. It is well known that field strain MG infections reduce table egg production in commercial egg-laying hens [4,5,6]. Carpenter et al. [1] and Evans and Hafez [7] have verified egg production losses in MG-infected layer chickens when compared to MG-free hens. In addition to a loss in egg production, egg quality can also deteriorate in hens infected with MG [3,4].

Though the frequency of MG infection has been drastically reduced over the last 50 years through eradication and isolation, it remains a current commercial concern [8]. In addition, a comprehensive review on the global occurrence of MG revealed that MG was more prevalent in broiler breeders than layers and that early detection and improved control measures are recommended to eliminate MG transmission [9]. Nevertheless, controlling MG infection via vaccines has been effective when isolation and biosecurity measures are impossible to achieve [4,10,11]. There are live vaccines that are used to manage the negative impacts of MG. Three of the MG vaccine strains (F strain of MG (FMG), 6/85 strain of MG (6/85MG), and ts-11strain of MG (ts-11MG)) that are currently commercially available have different features that can be useful in determining how they might be used to effectively control and eradicate field strains of MG. Depending on the type of infection being addressed, one vaccine strain might be found to be more effective than another. These features include protection [12], transmission [13], pathogenicity [7,14], and the ability to displace field strains [6,15].

Horizontal transmission of MG occurs through either direct or indirect contact with respiratory fluids, with the respiratory tract and conjunctiva serving as portals of entry into the recipient [8]. Hens can also transmit the infection vertically to offspring via transovarial or in ovo routes [8]. The FMG vaccine in particular is known to create higher levels of protection [12] as well as displacing the R_low_-strain of MG (RMG) [6]. Although the FMG vaccine provides more protection from field-strain MG infections, it is more virulent in comparison to the 6/85MG and ts-11MG vaccines and can negatively impact non-target poultry species. The 6/85MG and ts-11MG vaccine strains may be less effective, but are considered safer to use, as they are both apathogenic strains [6,7,12,14]. These differences are important to consider when attempting to formulate a plan for addressing the protection of a flock against field strain MG infections.

A series of studies were previously conducted to evaluate effects of the in ovo injection of FMG at 18 days of incubation (doi) on layer chick hatchability [16], posthatch survivability through 3 weeks and immunity through 6 weeks [17], and possible posthatch horizontal transmission through 12 weeks [18]. A 1 × 10^2^ colony forming units (cfu) or higher dosage of FMG delivered by in ovo injection resulted in higher embryonic [16,19] and posthatch [17] mortalities in comparison to noninjected or diluent-injected control groups. Elliott et al. [16] also observed that moderate (2.4 × 10^4^ cfu) and high (2.4 × 10^6^ cfu) doses of FMG significantly depressed posthatch growth. Although the use of a very low dosage (2.4 cfu) of injected FMG did not adversely affect chicks during the early posthatch period, it resulted in only a low humoral immune response [17]. Furthermore, Elliott et al. [18] reported that when FMG was applied in ovo at 18 doi, it was subsequently transmitted from vaccinated to unvaccinated birds when they were in direct contact, and Ley et al. [13] has confirmed that FMG can spread horizontally within a flock.

The live attenuated ts-11MG vaccine (ts-11MGV), which is licensed by USDA and is currently administered via eye-drop during the pullet phase, has been proven to be avirulent in chickens. The ts-11MGV is a live attenuated MG vaccine and possesses a temperature sensitive mutation generated by chemical mutagenesis of a moderately virulent Australian field isolate (strain 80083). The optimum temperature for its proper growth is 33 °C; however, ts-11MG growth has been shown to decrease at 39.5 °C [14]. A single dose of ts-11MG administrated by eye-drop application has been shown to result in colonization of this strain in the upper respiratory tract and to subsequently stimulate long-term immunity [20]. It is well documented that ts-11MG provides protection against respiratory disease and egg production drops induced by virulent MG [10,12,20] as well as exhibiting no egg transmission [21]. Vance et al. [22] has likewise shown that layers vaccinated with ts-11MG have exhibited no negative effects on egg production and egg size during the laying period.

In consideration of the adverse effects of in ovo-injected FMG on layer hatchability and posthatch growth, as noted by Elliott et al. [16,17], the objectives of this study were to examine the suitability of the delivery of ts-11MG via in ovo vaccination to layer embryos, its transmissibility after transfer to the hatcher and during the posthatch period, and its subsequent effects on various hatch variables and serologic responses.

## 2. Materials and Methods

### 2.1. Egg Incubation

A total of 2160 fertile Hy-Line W-36 layer hatching eggs were obtained from a 40 week-old commercial breeder flock (Hy-Line Company, Mansfield, GA, USA) that was deemed MG-clean by National Poultry Improvement Plan (NPIP) testing. Eggs were incubated in calibrated single stage NMC2000 NatureForm Incubators (Pas Reform North America, LLC, Jacksonville, FL, USA). In each of the 12 complete blocks, 30 eggs were set on each of six randomly arranged flats, with each flat representing an individual treatment. Each block occupied two tray levels in each of two columns of a single-stage incubator. At 18 doi, eggs in four of the treatment groups received diluent (Poulvac Marek’s disease diluent (Zoetis, Parsippany, NJ, USA)) containing one of four levels of the ts-11MGV, whereas two of the treatment groups were diluent-injected (sham-injected) control eggs, with one group being transferred on day of injection to a separate identical incubator to represent ts-11 MGV unexposed control (UC) eggs, while those that remained represented ts-11MGV exposed control (EC) eggs. The UC eggs were incubated alone in a separate incubator (designated as UC incubator) from the ts-11MGV-injected eggs to prevent possible ts-11MG cross-contamination. Thirty eggs were set on each of three flats on each of two tray levels in each of two tray columns (360 total eggs) of the UC incubator. The UC eggs were used for observational (non-statistical) comparison purposes for the base line values of eggs that had no potential exposure to the ts-11MGV during incubation. The eggs in the EC treatment were set in the same incubator (designated as EC incubator) with the ts-11MGV treatment groups for the detection of possible ts-11MG horizontal transmission between vaccinated and unvaccinated eggs or hatchlings during incubation. All eggs were incubated in a single incubator set at 37.50 °C dry bulb and 29.44 °C wet bulb (55% relative humidity) temperatures for the first 18 doi. The UC and EC incubators were in the same room, and both were set at a 36.70 °C dry bulb and 28.75 °C wet bulb (55% relative humidity) temperatures for the last four doi.

### 2.2. Treatment Designation and Application

At 12 and 18 doi, infertile eggs and those containing dead embryos were removed, and only those eggs containing viable embryos in each replicate treatment group received manual amniotic injections of diluent alone, or diluent containing one of 4 dosages of ts-11MGV at 18 doi. The manual amniotic injections were performed immediately after candling at 18 doi in accordance with the procedures of Embrex, Inc. [23] and that were likewise used in an earlier experiment by Elliott et al. [16]. The delivery volume was 50 μL. The amniotic treatments included controls that received only diluent (UC and EC). The titer of the ts-11MGV (VaxSafe MG (Strain TS-11) (Rhone Ma Holdings Berhad, Selangor, Malaysia)) was determined to be 7.25 × 10^6^ cfu/mL by duplicate plating. Plating was performed on Frey’s Mycoplasma agar [24] and incubated at 32.0 °C to confirm vaccine viability and the actual dosage being delivered. The manufacturer’s 1 × dose of the frozen ready-to-use solution of ts-11MGV was thawed and diluted in the diluent to achieve 3.63 × 10^1^, 3.63 × 10^2^, 3.63 × 10^3^, and 3.63 × 10^4^ cfu, respectively, of ts-11MG/50 μL. The 3.63 × 10^4^ CFU dosage represented the intended full dose of the vaccine administered. The 4 dosage levels, representative of the 4 ts-11MGV treatments, were administered on the day of injection. All ts-11MGV dosages were prepared on the same day of injection.

Localization of injections were verified by injecting dye into one embryonated egg from each of the five treatment groups on each of the 12 replicate flats in the EC incubator (60 total eggs) and one egg from each of the 12 replicate flats in the UC incubator (12 total eggs). Therefore, a total of 72 eggs were injected with dye across all treatment groups. The eggs were injected with Coomassie brilliant blue R-250 dye (Genlantis, San Diego, CA, USA) and opened immediately following the injection process to locate the site of injection.

After injection of the live embryonated eggs from each treatment replicate group, they were randomly allocated to six sections within each of two hatching baskets on each of 5 tray levels (one treatment per level) for the hatching phase in the ts-11MGV incubator that coincided with the treatment replicate groups represented in the setter phase. Hatching baskets containing eggs that received the high (3.63 × 10^4^ cfu) ts-11MGV dosage were placed at the bottom of the hatcher, and the lower doses (3.63 × 10^1^, 10^2^, and 10^3^ cfu, in that consecutive order, beginning at the top portion of the hatcher) treatment groups were placed so as to eliminate cross-contamination via chick droppings. In the UC incubator, after injection of embryonated eggs, UC eggs were allocated to six sections within each of two hatching baskets on one tray level.

### 2.3. Percentage Incubational Egg Weight Loss and Hatch Variables

Mean percentage incubational egg weight loss (PEWL) was determined for each replicate group of eggs in each treatment (total weight/number of eggs) between 0 and 12, 12 and 18, and 0 and 18 doi according to the procedure of Peebles et al. [25]. Because eggs were weighed after candling at 12 and 18 doi, mean PEWL between 12 and 18 doi did not include eggs that contained early dead embryos or eggs that were infertile or contaminated. Egg residues were marked, counted, and subsequently opened for embryonic development stage confirmation in accordance with the procedure described by Ernst et al. [26]. Percentages of hatchability of injected live embryonated eggs (HI), and pre-pipped embryo, pipped embryo, and hatched chick mortalities were determined at 22 doi. Bird husbandry, handling, sampling, and euthanasia procedures were approved by a USDA-ARS Animal Care and Use Committee (Mississippi State, MS, USA). In each of the six treatment groups on each of the replicate tray levels, mean straight run hatchling body weight (BW) was determined, and one chick was randomly selected for additional analysis. Each chick (72 total chicks) was weighed, euthanized, and their choanal clefts were immediately swabbed for MG DNA analysis. Birds in the UC and EC, and 3.63 × 10^1^, 10^2^, 10^3^, and 10^4^ cfu dosage treatment groups were swabbed in that order.

### 2.4. Posthatch Bird Raising and Sampling

Hatched chicks were counted and weighed and only females were raised to 3 weeks of age in the posthatch phase of the study. Female chicks from each replicate tray level that belonged to either the UC, EC, 3.63 × 10^2^ cfu, 3.63 × 10^3^ cfu, or 3.63 × 10^4^ cfu ts-11MGV treatment groups were pooled (approximately 300 birds). Twenty chicks from each of the five treatment groups were randomly assigned to each of three replicate biological isolated units (BIU) that were randomly arranged in a common room. Each BIU measured 1.42 m × 0.65 m (0.92 m^2^). Stocking density was 0.04 m^2^ per bird to meet Hy-Line W-36 breeder pullet recommendations [27]. Further information concerning the design [28] and internal conditions [18] of the BIU have been reported. To assess possible posthatch horizontal transmission, five male sentinel chicks derived from the UC treatment were also allocated to each replicate BIU belonging to the EC and 3 ts-11MGV treatments. Due to space limitations, the lowest dosage treatment (3.63 × 10^1^ cfu) was not maintained beyond hatch. The birds belonging to that treatment group were weighed before being humanely euthanized. This lowest dosage treatment group was eliminated, as its hatch results were not statistically different from the other treatments.

Each BIU possessed an observation window and had a bell drinker and a circular gravity-fed feeder that would accommodate up to 10 to 12 birds simultaneously. A flat feed tray was provided for the first week posthatch before the feeder was used. All birds had ad libitum access to water and feed during the entire grow-out period. The pullets were fed a crumble starter diet that met or exceeded NRC requirements [29]. According to Hy-Line recommendations for W-36 pullets, the lighting program provided 21 h of light and 3 h of dark in the room during the first week, and 20 h of light and 4 h of dark until day 21 of age. Furthermore, as recommended, room air temperature was 34 °C on the first day and was lowered by 2 to 3 °C daily until 21 °C was reached. Birds belonging to the UC control treatment were monitored before those belonging to the EC control treatment. Subsequently, birds belonging to the EC treatment were monitored before those belonging to the ts-11MGV treatments to prevent possible cross-contamination. Birds in the 3.63 × 10^2^, 10^3^, and 10^4^ cfu treatment groups were likewise monitored in that respective order, to further prevent possible cross-contamination. Chick mortality was monitored daily with dead chicks being weighed daily. However, in order to further prevent the possibility of cross-contamination, if there were no mortalities or there was no need to add feed, the units were not opened. After chick placement in each replicate BIU, mean female chick BW was determined on day 0 (BW0) and 21 (BW21) posthatch. The difference between mean BW0 and BW21 was used to calculated mean 0 to 21 day posthatch BW gain (BWG21) for each replicate BIU. The UC sentinel birds were not included in the assessment of the posthatch performance variables.

### 2.5. Choanal Cleft Swabbing

Pre-wetted sterile swabs in phosphate-buffered saline were used to swab the choanal cleft of the birds at hatch and at 3 weeks posthatch for MG detection. All swabs were suspended in 100 μL of phosphate-buffered saline and extracted using a BioSprint 96 One-For-All Kit (Qiagen, Valencia, CA, USA). All swab samples were run in duplicate using a 7500 Fast Real Time PCR System (Applied Biosystems, Foster City, CA, USA). An MG-specific primer and probe set (Table 1) as described by Callison et al. [30] were used for detection of ts-11MG. Primers and probes were diluted to 0.5 μM and 0.1 μM, respectively. Reactions were conducted at 25 μL volumes, which included 2.5 μL of forward and reverse primers, 2.5 μL of probe, 12.5 μL of Taqman Universal Master Mix (Applied Biosystems, Foster City, CA, USA), and 2.5 μL of extracted DNA in sterile water [31]. A 22.5 μL volume of master mix was loaded into each well of a 96-well-plate, and 2.5 μL of sample DNA was then subsequently added to each well. The reaction was performed as described by Callison et al. [30] on a 7500 Fast Real Time PCR system (Applied Biosystems, Foster City, CA, USA). Briefly, samples were incubated at 50 °C for 2 min and 95 °C for 15 min with optics off followed by 40 cycles of 94 °C for 15 s and 60 °C for 1 min. Each plate included a negative control that contained sterile water only, and a positive control containing RMG to design a standard curve including 10^−1^, 10^−2^, and 10^−3^ dilutions.

### 2.6. Blood Sampling and Immunology

After the birds were swabbed at 3 weeks of age, they were immediately bled and approximately 3 mL of blood from each bird was collected and kept at room temperature for 2 h. After clotting, sera were centrifuged and poured into labelled vials and stored at 4 °C until use. For detection of IgM antibodies against MG, SPA analysis was conducted according to the procedures described by Kleven [32]. Serum plate agglutination testing with MG antigen (Charles River Laboratories International, Wilmington, MA, USA) was applied using the 0 to 3 scale according to the procedure of Evans et al. [33]. Scores of 1 or higher were considered positive SPA tests.

### 2.7. Statistical Analysis

A randomized complete block design was employed for PEWL and the hatch variables, which included HI, egg residue analyses, and hatchling BW, where a group of 30 eggs in each flat represented an experimental unit, and where two tray levels in each of two tray columns were considered as the blocking factor. All treatments were randomly assigned to each block. Those treatments housed in the same incubator, and that were exposed to ts-11MGV (EC and ts-11MGV treatment groups) were compared statistically using two-way ANOVA. The UC control treatment was not analyzed with the EC control and ts-11MGV treatments and was only used for numeric comparison. Employing one-way ANOVA, posthatch BW0, BW21, and BWG21; MG DNA detection via Fast Real Time PCR; and MG antibodies detected by SPA were analyzed using a completely randomized experimental design, where replicate BIU was the experimental unit. All data were analyzed using SAS 9.4 [34] employing PROC MIXED, and means separations were performed using Fisher’s protected least significant difference in the event of significant global effects. Statements of significance were based on *p* ≤ 0.05 unless otherwise stated.

## 3. Results

### 3.1. Localization of Injections

The dye injection results confirmed that except for two eggs in one of the replicate-treatment-groups, all of the eggs in the replicate-treatment groups received amnion (AM) injections. The 3.63 × 10^4^ cfu treatment was the exception, in which 10 of the 12 (83.3%) dye-injected eggs received AM injections, whereas embryos in 2 of the 12 (16.7%) eggs received intramuscular injections. Of those eggs injected with dye, 97.2% of the injections across all treatment groups were in the AM, and 2.8% were intramuscular.

### 3.2. Prehatch, Hatch, and Posthatch Variables of the Unexposed Diluent-Injected Control (UC) Treatment

The swab samples from all the birds in all treatment groups at hatch and at day 21 posthatch possessed no MG DNA that was tested by Real Time PCR. All birds in all treatment groups also tested SPA negative for IgM on day 21 posthatch. Further analysis by ELISA for the presence of IgG antibodies against MG was to be conducted should SPA analysis have revealed any positive samples. However, ELISA tests were not subsequently performed due to the absence of any SPA positive results. In the UC treatment, mean 0 to 12 doi PEWL, 12 to 18 doi PEWL, 0 to 18 doi PEWL, HI, hatchling BW, and pre-pipped embryo, pipped embryo, and hatched chick mortalities were 7.23%, 3.92%, 11.15%, 96.63%, 40.32 g, 0.98%, 0.32%, and 0.99% respectively. The PEWL, HI, and hatchling BW (Table 2); and pre-pipped embryo, pipped embryo, and hatched chick mortality (Table 3) results in the EC incubator were comparable to those in the UC incubator. The UC treatment group means for BW0, BW21, and BWG21, were 40.2 g, 192.1 g, and 151.9 g, respectively.

### 3.3. Prehatch, Hatch, and Posthatch Variables of the Exposed Diluent-Injected Control (EC) and Ts-11MGV Treatments

The EC treatment group occupied the same incubator as those injected with the various ts-11MGV doses to test for ts-11MG transmission during the incubational period. For the EC control and 3 ts-11MGV treatments, the swab samples possessed no MG DNA at hatch. Thus, there was no evidence of ts-11MG transmission from in ovo-vaccinated to unvaccinated chicks within the same hatcher unit. Furthermore, at day 21 posthatch, swab samples possessed no MG DNA, and there were no IgM serological responses (negative SPA tests) at day 21 posthatch due to the delivery of the ts-11MGV via in ovo injection. The lack of evidence of ts-11MG transmission to the EC treatment within the hatcher suggests that the EC group remained MG-free and were truly a clean control group for comparison to the ts-11MGV treatment groups. The prehatch and hatch variables in the EC and 4 ts-11MGV treatment groups (3.63 × 10^1^, 3.63 × 10^2^, 3.63 × 10^3^, and 3.63 × 10^4^ cfu) were statistically compared. Mean 0 to 12 doi PEWL, 12 to 18 doi PEWL, 0 to 18 doi PEWL, HI, and hatchling BW were not significantly different between these five treatment groups (Table 2). In addition, there were no significant differences between the five treatment groups for pre-pipped embryo, pipped embryo, and hatched chick mortalities (Table 3). These results indicate that all treatment groups experienced similar incubational conditions. These combined results also suggest that injection of the ts-11MGV at 18 doi has no negative impact on embryo development and subsequent hatch variables. Furthermore, when incubated in the same incubator, there is no observable horizontal transmission of ts-11MG between hatchlings.

There were no posthatch mortalities recorded between days 1 and 21 posthatch in any of the replicate pens within each treatment group. There were no significant differences among the EC and 3 ts-11MGV treatment groups for BW0 and BW21 (Table 4). However, BWG21 in the 3.63 × 10^4^ cfu treatment group was significantly (*p* = 0.023) lower than that in the EC, 3.63 × 10^2^ cfu, and 3.63 × 10^3^ cfu treatment groups (Table 4). The BW0, BW21, and BWG21 results of the UC group were comparable to those in the EC group, which further establishes that the EC treatment is likewise sufficient for statistical control comparison to the ts-11MGV treatment groups during the posthatch phase.

## 4. Discussion

Wakenell et al. [35] has reported that optimal (90% protective index) efficacy of the Marek’s disease vaccine was found when injected via amniotic or intraembryonic routes. Therefore, the dye injection results of this study showing 97.2% AM and 2.8% intramuscular injections across all treatments would indicate that the ts-11MGV was successfully administered in the injected eggs. However, no morbidity was detected in the pullets and the male sentinel birds in each treatment group, and their swab and blood samples were negative for MG presence and IgM antibody production at day 21 posthatch. This contrasts to the results of a study conducted by Elliott et al. [18], in which layer pullets that were vaccinated with an FMG vaccine by in ovo injection, were shown to be capable of transmitting FMG to other pen mates that they were in direct contact with. In this study, birds that hatched from eggs in the EC treatment remained ts-11MG free through 21 days posthatch, and ts-11MGV was not transmitted to male sentinel birds that were placed in contact with the female birds belonging to the 3 ts-11MGV treatments. Ley et al. [13] likewise reported that ts-11MG is relatively avirulent and when administered during the pullet phase, exhibited no transmission from ts-11MGV vaccinates to unvaccinated sentinel birds that occupied adjoining pens and that had indirect contact with the vaccinates. Furthermore, no birds displayed any clinical signs of morbidity or mortality, and no gross lesions were observed when necropsied [13]. Kleven et al. [6] also reported that there is virtually no bird-to-bird transmission in response to the posthatch administration of ts-11MGV. Nevertheless, this study has further revealed that ts-11MG exhibits no successive horizontal transmission when administered by in ovo injection. Furthermore, differences in the results of the ts-11MG and FMG studies can be attributed to the much lower virulence and greater environmental sensitivity of ts-11MG.

The objectives of the current study also included determinations as to the effects of the in ovo injection of ts-11MG on the serologic responses, and on various hatch and posthatch performance variables of Hy-Line W-36 layers. The results of this study showed that in ovo vaccination of ts-11MG had no negative effects on the hatch variables examined, and that doses up to 3.63 × 10^3^ cfu showed no outward clinical indications of any negative physiological impacts on the embryos. No negative effects of the 3.63 × 10^3^ cfu dosage on subsequent posthatch BW or BW gain were also noted. The in ovo inoculation of ts-11MGV and its subsequent effects on transmissibility, hatch quality, and posthatch live performance have not been previously reported. However, these results agree with those of Vance et al. [22], in which pullets that received eye drop vaccinations of 1 × 10^6^ cfu of ts-11MG at 10 weeks of posthatch age experienced no negative effects on their BW between 18 and 57 weeks of posthatch age. Apparently, a 10^6^ cfu dosage of the ts-11MGV given prelay of 10 weeks of age [22] causes no physiological consequences in the pullet period. Biro et al. [36] and Whithear et al. [14] also observed that after hens were vaccinated with ts-11MG by eye drop at 22 weeks of age, no evidence of ts-11MG was detected in their eggs or oviducts. Nevertheless, the highest dosage used in this study (3.63 × 10^4^ cfu) led to a decrease in BWG21, which suggests that a successful vaccination process may have existed and that when administered by in ovo injection, ts-11MGV may cause a subsequent negative impact on posthatch live performance, whereas when given to pullets at doses as high as 10^6^ cfu, no adverse effects on live performance are realized.

After Elliott et al. [16] injected FMG into the AM, it was detected in the trachea, mouth, esophagus, yolk sac, and lumen of the duodenal loop at 22 doi, but hatch of embryonated eggs was decreased by the injection of 1 × 10^6^ and 1 × 10^4^ cfu dosages of FMG. Only a very low dosage between 5.06 and 5.93 cfu gave an acceptable level of hatch that averaged 95.8%. In a subsequent study by Elliott et al. [16], it was reported that in ovo injection of 2.4 cfu of FMG had the least posthatch impact on livability through 3 weeks and on BW through 6 weeks, but it exhibited the lowest humoral immune response at 6 weeks. It was further shown in that report that the 2.4 × 10^4^ and 2.4 × 10^6^ cfu in ovo doses of FMG that were applied by in ovo injection at 18 doi resulted in more than a 50% posthatch chick mortality. Elliott et al. [17] also observed that the posthatch BW of the birds at 6 weeks of age belonging to the 2.4 × 10^2^, 10^4^, and 10^6^ cfu FMG in ovo treatment groups were significantly lower compared to those given the lower dosages and to those in a diluent-injected control group. In addition, only the very lowest dosage of FMG (2.4 cfu) allowed for a 95.2% hatchability and normal posthatch growth, whereas in the current study, up to a 3.63 × 10^4^ cfu dosage showed no negative impact on hatchability.

However, when layers were inoculated with 10^6^ cfu of ts-11MG at 10 weeks of age alone or in conjunction with an overlay inoculation of FMG at 45 weeks of age, their live performance was not adversely affected [22]. It appears that up to a 3.63 × 10^4^ cfu dosage of ts-11MG is tolerated when it rather than FMG is used for in ovo injection. This can be attributed to the lower virulence level of ts-11MG [37]. Furthermore, when FMG was applied in ovo at 18 doi, it was successfully transmitted from vaccinated to clean sentinel chicks when they were in direct contact [18]. Nevertheless, in this study, no posthatch ts-11MG transmission occurred between vaccinated and clean sentinel birds that occupied the same BIU regardless of the ts-11MGV dosage used. The lack of horizontal transmission in this study is associated with the lack of any observable MG DNA or IgM serological responses in the chicks that hatched from eggs that were administered the ts-11MGV by in ovo injection.

In the current study, the ts-11MGV was administrated at 18 doi when embryos have an immature immune system, and it is well documented that at least 2 to 3 weeks are required for lymphocytes to be fully developed in posthatch chickens [38]. Moreover, ts-11MG is not detected systemically in birds until 2 to 3 weeks post vaccination [10]. Serum antibody responses in chickens vaccinated with ts-11MG have been shown to be variable [39], and ts-11MG can be difficult to detect serologically, as low antibody levels that are slow to develop occur in response to a ts-11 MG vaccination [8]. These findings might provide an explanation as to why ts-11MG was not detected at hatch and had no effects on the hatch variables examined in the current study. Layer embryos may be more sensitive than pullets to higher dosages of ts-11MG, and dosages of 3.63 × 10^4^ cfu or greater can have subsequent adverse effects on posthatch growth when administered via in ovo injection. Although the duration of the posthatch phase of the current study was 3 weeks, further effects of the ts-11MGV may have been realized should the study have been extended for a longer period of time. Nevertheless, the results of the current study are largely in agreement with these earlier studies, and depending on the dosage delivered, indicate that bird growth may not necessarily be adversely affected whether ts-11MG is administered by in ovo injection or later in the prelay pullet period.

The birds in this study did not show an immune response to the ts-11MGV treatments imposed. Furthermore, for all treated birds, no MG DNA was detected in their choanal clefts at hatch or 21 days posthatch, and their SPA tests for IgM were negative at 21 days posthatch. Although the level of ts-11MG present in the birds may have been below the threshold of detection by the PCR test employed, successful growth was observed in all the injected solutions that were plated, which validated the functionality of the injected solutions containing the various dosage of ts-11MGV. This precluded the possibility that structural changes in the ts-11MGV might have occurred after its in ovo injection on 18 doi. However, the high incubation temperatures may have also affected the efficiency of the injected solutions. The efficiency and functionality of ts-11MG is reduced at 39.5 °C [14]. In addition, eggshell temperature, which is an indicator of embryonic temperature, is approximately 38 °C from 18 to 21 doi [40]. Furthermore, ts-11MG is a mutant strain that has been shown to lack certain genes which can result in a reduction in its functionality [41,42]. The GapA gene is the primary cytadhesin gene for MG [41,42], and MG attaches to target cells via cytadhesins where interactions occur with their corresponding host cells receptors [43,44]. The GapA gene is expressed in virulent MG strains as well as in FMG, where it is important for cytadherence and their subsequent virulence properties [45]. However, the GapA gene is absent in the ts-11MG vaccine strain [46], which may make the ts-11MG vaccine less efficient. Studies have found that GapA and CrmA gene co-expression are associated with MG cytadherence and virulence [45]. The absence of the GapA gene, but not the CrmA gene, is identifiable in the ts-11MG vaccine strain [46], and the expression of the CrmA gene is not dependent on expression of the GapA gene [47]. In addition, the release of chicken interleukin-6 by ts-11MG C3 (+CS) has been shown to be positively linked to BW gain [48]. In layers in ovo-injected with ts-11 MGV, further study is needed to determine their serological responses for periods longer later than 3 weeks, and to identify the genes that might have been missing when ts-11 MGV was administrated in ovo.

## 5. Conclusions

In conclusion, the in ovo injection of ts-11MGV at 18 doi had no negative impact on the HI and posthatch serology of layers. However, it is possible that physiological (organ and blood) and immunological variables that were not investigated in this study may have played a role in the negative effects that the in ovo injection of ts-11MGV had on BWG21. Although ts-11MG is relatively avirulent in comparison to other vaccine strains of MG such as FMG, the embryo appears to be more sensitive to ts-11MG when applied in ovo rather than as an eye drop in the pullet phase. This higher level of sensitivity could be related to the immaturity of the immune system of late-stage embryos. This was evidenced by the decrease in BWG21 in the group that received the highest dosage. This delayed effect on BWG21 may be because ts-11MGV requires a longer time for systemic colonization due to its low level of virulence. Furthermore, the in ovo administration of ts-11MG was not transmitted from vaccinated to sentinel birds at both the hatch and posthatch periods and no subsequent MG DNA or serology responses were detected. However, further research is required to measure serological responses to in ovo-injected ts-11 MGV for a longer time interval, and to incorporate a challenge model. The undetectable antibody levels against MG at 3 weeks posthatch may result in a weak vaccination consistency among all birds in a flock when ts-11MGV is applied in ovo. Because transmission of an MG vaccine between birds in a flock can assist in total flock protection, this is a matter of concern for the in ovo use of ts-11MGV. Therefore, ts-11MGV may not be a good candidate for in ovo injection for the subsequent protection of commercial layer flocks against field-strain MG infections.

## Figures and Tables

**Table 1 animals-12-01120-t001:** Primers and probe used for DNA analysis.

Item		Base Sequence
Primer	mg1pU26	5′-CTA gAg ggT Tgg ACA GTT ATg-3′
Primer	mg1p164	5′-gCT gCA CTA AAT gAT ACg TCA AA-3′
Probe	mg1pprobeA	5′-6FAM-CAg TCA TTA ACA ACT TAC CAC CAg AAT CTg-BHQ_1-3′

**Table 2 animals-12-01120-t002:** Mean percentage egg weight loss (PEWL) in the 0 to 12, 12 to 18, and 0 to 18 days of incubation (doi) intervals, hatchability of injected eggs containing viable embryos (HI), and mean straight run hatchling BW at 22 doi in ts-11MG exposed diluent-injected control (EC) eggs and those injected with 3.63 × 10^1^, 3.63 × 10^2^, 3.63 × 10^3^, and 3.63 × 10^4^ cfu of the ts-11MG vaccine (ts-11MGV).

Treatment	PEWL0 to 12 doi ^6^	PEWL12 to 18 doi	PEWL 0 to 18 doi	HI ^7^	Hatchling BW ^8^
	(%)				(g)
EC ^1^	7.09	3.94	11.08	97.24	39.81
3.63 × 10^1^ cfu ^2^	7.09	3.59	10.67	97.48	39.70
3.63 × 10^2^ cfu ^3^	7.15	4.00	11.12	96.15	40.00
3.63 × 10^3^ cfu ^4^	7.13	4.15	11.29	96.43	39.94
3.63 × 10^4^ cfu ^5^	7.06	4.01	11.08	97.93	40.16
Source of Variation					
Pooled SEM	0.090	0.196	0.207	1.463	0.299
*p*-value	0.856	0.080	0.063	0.727	0.595

^1^ At 18 doi, embryonated eggs injected with Marek’s diluent were incubated in the same hatcher with ts-11MGV injected eggs. ^2^ 10^1^ cfu (colony forming unit) of ts-11MG was a 50 μL volume of a 10^−4^ dilution from a resuspended vial of vaccine. ^3^ 10^2^ cfu of ts-11MGV was a 50 μL volume of a 10^−3^ dilution from a resuspended vial of vaccine. ^4^ 10^3^ cfu of ts-11MGV was a 50 μL volume of a 10^−2^ dilution from a resuspended vial of vaccine. ^5^ 10^4^ cfu of ts-11MGV was a 50 μL volume of a 10^−1^ dilution from a resuspended vial of vaccine. ^6^ PEWL was calculated based on the difference between the initial and final weight of 30 eggs in each of 12 replicate flats in each (0 to 12), (12 to 18), and (0 to 18) doi period. ^7^ Average percentage of hatchability of viable injected eggs containing embryos that hatched successfully and were alive at pull time. ^8^ Hatchling BW was calculated from approximately 20 chicks in each of 12 replicate basket sections in each treatment.

**Table 3 animals-12-01120-t003:** Mean pre-pipped embryo, pipped embryo, and hatched chick mortality at 22 days of incubation (doi) in exposed diluent-injected control (EC) eggs and those injected with 3.63 × 10^1^, 3.63 × 10^2^, 3.63 × 10^3^, and 3.63 × 10^4^ cfu of the ts-11MG vaccine (ts-11MGV).

Treatment	Pre-Pipped Embryo Mortality ^2^	Pipped Embryo Mortality ^3^	Hatched Chick Mortality ^4^
	(%)		
EC ^1^	0.00	0.00	2.92
3.63 × 10^1^ cfu	0.33	0.00	2.06
3.63 × 10^2^ cfu	0.29	0.36	3.18
3.63 × 10^3^ cfu	0.00	0.33	2.86
3.63 × 10^4^ cfu	0.00	0.00	0.94
Source of Variation			
Pooled SEM	0.204	0.215	0.988
*p*-value	0.577	0.568	0.501

^1^ At 18 doi, embryonated eggs injected with Marek’s diluent were incubated in the same hatcher with ts-11MGV injected eggs. ^2^ Embryos externally pipped through the eggshell that were alive but had not fully hatched at pull time at 22 doi. ^3^ Embryos that had externally pipped but were dead at pull time. ^4^ Fully hatched chicks that were found dead in the hatch basket at pull time.

**Table 4 animals-12-01120-t004:** Mean BW at 0 (BW0) and 21 (BW21) days posthatch, and BW gain between 0 and 21 days posthatch (BWG21) in exposed diluent-injected control (EC) eggs and those injected with 3.63 × 10^2^, 3.63 × 10^3^, and 3.63 × 10^4^ cfu of the ts-11MG vaccine (ts-11MGV).

Treatment	BW0 ^5^	BW21 ^5^	BWG21 ^5^
	(g)		
EC ^1^	39.60	183.67	144.00 ^a^
3.63 × 10^2^ cfu ^2^	39.50	185.67	146.00 ^a^
3.63 × 10^3^ cfu ^3^	40.57	185.67	145.33 ^a^
3.63 × 10^4^ cfu ^4^	40.63	176.67	136.67 ^b^
Source of Variation			
Pooled SEM	0.434	2.45	2.33
*p*-value	0.073	0.058	0.023

^a,b^ Treatment means within the same variable column with no common superscript differ significantly (*p* ≤ 0.05). ^1^ At 18 doi, embryonated eggs injected with Marek’s diluent were incubated in the same hatcher with ts-11MGV injected eggs. ^2^ 10^2^ cfu of ts-11MGV was a 50 μL volume of a 10^−3^ dilution from a resuspended vial of vaccine. ^3^ 10^3^ cfu of ts-11MGV was a 50 μL volume of a 10^−2^ dilution from a resuspended vial of vaccine. ^4^ 10^4^ cfu of ts-11MGV was a 50 μL volume of a 10^−1^ dilution from a resuspended vial of vaccine. ^5^ BW0, BW21, and BWG21.

## Data Availability

None of the data were deposited in an official repository.

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
