# Peer review of "Effects of the In Ovo Vaccination of the ts-11 Strain of Mycoplasma gallisepticum in Layer Embryos and Posthatch Chicks†"

_animals, 2022, doi:10.3390/ani12091120_

Round 1

Reviewer 1 Report

Please consider changing the title of the study by eliminating the words “ transmission and impact and temperature-sensitive” and instead include “effect of in ovo-application of ts-11MGV Mycoplasma gallisepticum live vaccine.. ”

“Though the frequency of MG infection has been drastically reduced over the last 50 years through eradication and isolation, it remains a current commercial concern” Following these lines, I recommend including some results regarding the global molecular prevalence estimations for MG presented in the following study: Chaidez-Ibarra MA, Velazquez DZ, Enriquez-Verdugo I, et al. Pooled molecular occurrence of Mycoplasma gallisepticum and Mycoplasma synoviae in poultry: A systematic review and meta-analysis. Transboundary and Emerging Diseases 2021. https://doi.org/10.1111/tbed.14302

“Three of the MG vaccine strains [F strain of MG (FMG), 6/85 strain of MG (6/85MG), and ts-11strain of MG (ts-11MG)] that are currently commercially available have different features” …” These features include protection [11], transmission [12], pathogenicity [7,13], and the ability to displace field strains [6,14]. ” Please provide specific details regarding the differences and similitudes among the vaccines. Maybe this essential information could be organized in a table. Authors could also include some of the improtatn differences presented in lines 78-83.

“Depending on the type of infection being addressed, one vaccine strain might be found to be more effective than another. ” Please elaborate more on these differences because this is important to establish the context of the present study.

Despite the authors present an important series of results regarding the use of FMG as an in ovo injection in lines 86-97, I was not able to understand the rationale for using ts-11MGV for in ovo vaccination. The fact that FMG causes adverse effects when applied in ovo does not directly justify the use of ts-11MGV applied in this way.  Besides, the effect of ts-11MGV has been previously shown. To provide a better rationale, I recommend that the authors should include a better and comparative context regarding vaccination schemes and plans for controlling and eradicating the infection. I mean, explain why it is important to apply an in ovo MG vaccine.  

Statistical analysis. Please specify how did the authors perform the random assignment. Fisher LSD is an outdated multiple comparison test, besides this test does not correct for multiple comparison tests. Therefore, I recommend authors use instead Tukey HSD or Bonferroni as the post-hoc test. Please read the following: https://www.graphpad.com/guides/prism/latest/statistics/stat_fishers_lsd.htm

https://www.graphpad.com/guides/prism/latest/statistics/stat_how_the_fisher_lsd_method_work.htm

For all statistical results, please provide numerical values of statistical significance (for ANOVA F, df, exact p-values) of the main effects and interactions for each data set when available. All statements concerning statistical significance must be qualified numerically. It is not sufficient to provide p-values only.

Reviewer 2 Report

This MS described "the transmission and impact of an in ovo-applied temperature-sensitive Mycoplasma gallisepticum line vaccine in layer embryos and posthatch chicks". This appeared to be a good study with the appropriate design and worthy a merit of publication. However, there were still some trivial points needed to be fixed before it is acceptable. Therefore, please make necessary changes without any preservation. 

Specific comments were as following:

  1. Please clarify the conclusion of your abstract. Please add more information as to" .......it may not be a good candidate for in ovo injection.". Actually you did not brief why?
  2. Line 39. Wrong typos. Please correct 3.63x101 to 3.63x10^1..........
  3. Lines 100 to lines 115. Watch out the word-type of letters in your sentences. They were different from others.
  4. Lines 401 to lines 415. You spent the whole paragraph mentioning the work done by Elliott without any linkage to the present results of your study. I did not comprehend the meaning of including this section in your discussion. Please omit or reorganize this.
  5. lines 445 to lines 471. This part seemed very significant for me to give the readers a brilliant explanation about the outcome of this study. You better move this forward in your discussion. 
